

# Monitoring Groundwater Storage Depletion Using Gravity Recovery and Climate Experiment (GRACE) Data in the Semi-Arid Catchments

Nizar Abou Zaki[1], Ali Torabi Haghighi[1], Pekka M. Rossi[1], Mohammad J. Tourian[2], Bjørn Kløve[1].

[1]Water Resources and Environmental Engineering Research Group, University of Oulu, Oulu, Finland.

[2]Institute of Geodesy, University of Stuttgart, Stuttgart, Germany.

*Correspondence to*: Nizar Abou Zaki (nizar.abouzaki@oulu.fi)

## Abstract

The Bakhtegan catchment, an important agricultural region in south-western Iran, has suffered groundwater depletion in recent years. As groundwater is considered the main source of fresh water in the catchment, especially for agriculture, monitoring groundwater responses to irrigation is important. Gravity Recovery and Climate Experiment (GRACE) satellite data can help determine water mass changes in catchments and assess water volume changes, but have been under-used to date in water

resources management. In this study, we compared GRACE-derived water mass data against groundwater volume variations measured in situ. We also assessed the efficiency of GRACE-derived data in catchments smaller than the 200,000 km$^2$ recommended area when using GRACE. For the study period (January 2002 through December 2011), the GRACE data showed a 7.6 mm annual decline in groundwater level, with a total volume loss of 2.6 km$^3$ during the period. The in situ monthly

measurements of groundwater level showed an average depletion of 10 m in catchment aquifers during the study period. This depletion rate was supported by the recorded decrease in precipitation volume, especially in the post-drought period after 2007. These results demonstrate that GRACE can be useful in groundwater resources management of catchments facing groundwater depletion and increasing water demand.



## 1. Introduction

Increasing demand for water supply in arid and semi-arid zones has increased groundwater use, leading to wide-scale depletion (Scanlon et al., 2006). Semi-arid zones represent 30% of global terrestrial surface area and water scarcity in these regions is a severe problem, due to rapid population growth and

expansion of irrigated agriculture (Dregne, 1991). In arid and semi-arid areas where surface water resources are scarce and unreliable, groundwater is considered the only plausible fresh water source, due to its quantitative and spatial availability (MacDonald et al., 2012). Many of the groundwater resources developed in arid and semi-arid zones are non-renewable fossil water (Sultan et al., 2007). Sustainable management of aquifers in arid and semi-arid zones requires accurate estimates of recharge

rate and groundwater resources, data which are often lacking in developing countries.

Gravity Recovery and Climate Experiment (GRACE) satellite data are a valuable resource for water storage monitoring (Famiglietti et al., 2011). GRACE provides temporal and spatial records of total water mass storage variations, including snow, surface water, soil moisture, and groundwater (Voss et al., 2013). These records can help forecast effects such as groundwater depletion, desertification, and

changes in surface water bodies (Rodell and Famiglietti, 2009). More importantly, GRACE can provide data for regions that lack conventional hydrological monitoring infrastructure. Groundwater level measurements have already been compared with GRACE data for the Tigris-Euphrates basin (Voss et al., 2013), Lake Urmia in Iran (Tourian et al., 2015), the California Valley (Scanlon et al., 2012), Northern China (Feng et al., 2013), and Northern India (Rodell et al., 2009).

In this study, we compared GRACE-derived water mass data against in situ monthly measurements of groundwater level obtained from 448 observation wells distributed throughout the semi-arid Bakhtegan catchment. We compared water mass storage variations for 117 months, from April 2002 until December 2011. Moreover, we analyzed precipitation and evapotranspiration data, in order to characterize net precipitation. The aim of the study was to test the viability of applying GRACE-derived

data together with in situ measurements to understand the hydrological cycle in a vulnerable catchment showing a recent drastic decline in groundwater resources.



## 2. Materials and Methods

### 2.1 Study Area

The Bakhtegan catchment is located in the north-eastern part of Fars province in south-western Iran (Figure 1). The total catchment area is 31 511 km$^2$ and includes 16 630 km$^2$ of mountainous area and 14 881 km$^2$ of plains (Hedayat et al., 2017). The Kor River, the main river in the catchment, originates in the Zagros Mountains and is 280 km long (Haghighi and Kløve, 2017). Bakhtegan and Tashk are the main lakes in the catchment, with a combined area which ranged between 220 and 640 km$^2$ during the study period. Mean long-term annual precipitation in the catchment is 270 mm (for the 43-year period 1967-2009) (Choubin et al., 2016). Thus most areas in the Bakhtegan catchment have an arid or semi-arid climate (Hojjati and Boustani, 2010). The catchment is divided into 27 separate groundwater monitoring zones based on alluvial aquifer distribution, with 448 observation wells recording changes in groundwater level. The alluvial aquifers have a total area of 10 564 km$^2$, with average aquifer thickness ranging between 30 and 50 m (Rasoulzadeh and Moosavi, 2007). The alluvial sediments consists of rubble, stone, gravel, sand, and silt, with low amounts of clay, and most of the aquifers are situated in valleys between highlands (Figure 1).

As in all arid and semi-arid regions, groundwater is fundamental for social and economic stability in Fars province (Soltani et al., 2008). The good availability of groundwater to date, the low exploitation cost, and the relatively good quality have made it the most reliable and useful water source in the region. However, due to lack of surface water resources in the Bakhtegan catchment, groundwater extraction has exceeded the renewable recharge limits in recent years, with significant effects on surface water resources (Hedayat et al., 2017). This excessive groundwater exploitation, in combination with recent droughts and lack of alternative surface water resources, has also led to depletion of groundwater levels (Haghighi and Keshtkaran, 2008). About 4 billion m$^3$ of groundwater are extracted annually from 40 000 wells and boreholes in Fars province (Hojjati and Boustani, 2010). With expected population growth, climate change, and increased groundwater extraction, sustainable groundwater management is essential in conserving the future sustainability of groundwater resources in the region.



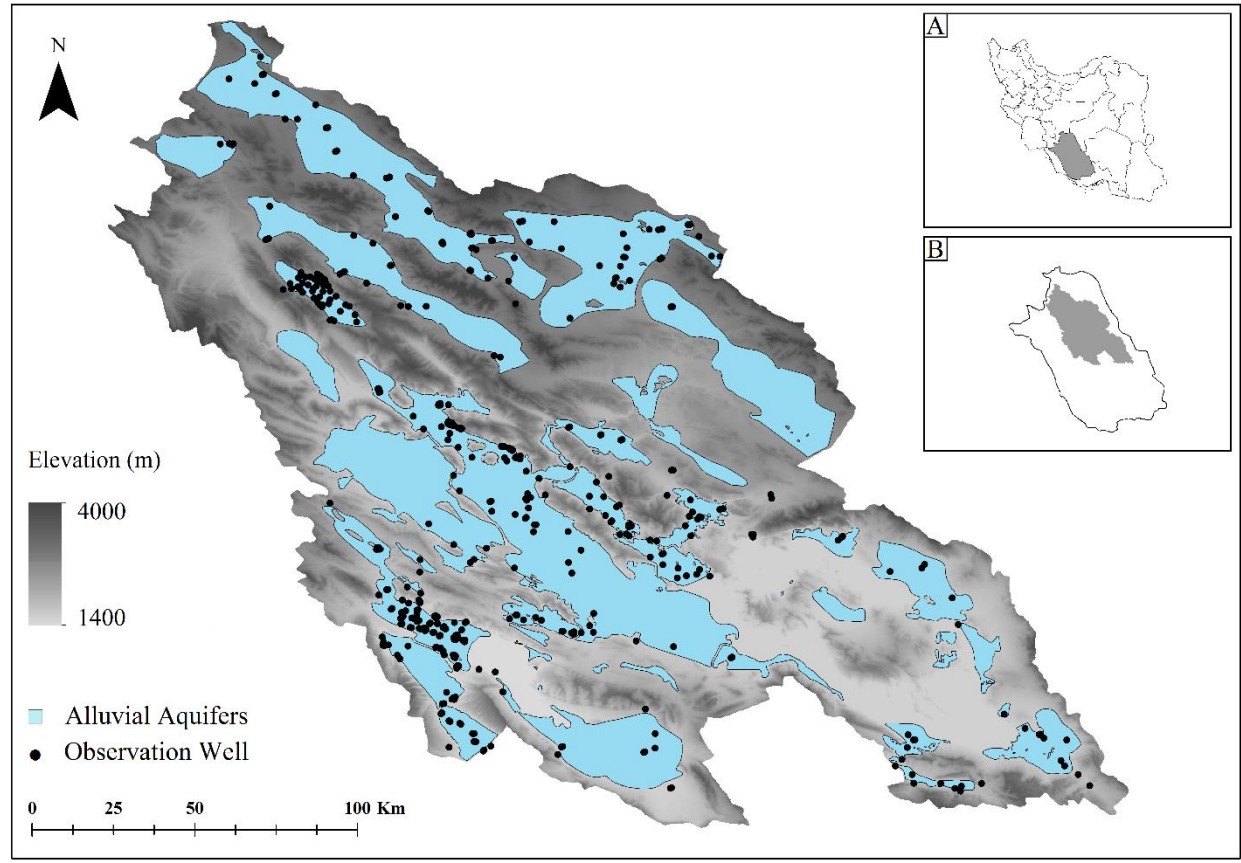

**Figure 1**: *Distribution of alluvial aquifers and observation wells in Bakhtegan catchment. Inserts: Location of [A] Fars province in Iran and [B] the Bakhtegan catchment in Fars province.*

## 2.2 Calculating Net Precipitation

Data on decadal variations in precipitation from April 2002 to December 2011 were obtained from 22 local observation stations (Water Resources Atlas Report, 2011). The interpolated values show variations due to elevation (which ranges from 1420 to 2260 m above sea level) (Figure 2). Land use data and average evapotranspiration rates for the Bakhtegan catchment (USGS MODIS, clim-engine.appspot.com) were used to estimate the rate of evapotranspiration from agricultural land, forests,





and lakes in the catchment. Data on observed monthly precipitation volume, land use, and evapotranspiration rates were then combined to calculate net precipitation:

$$W_B = P - E_T \qquad (1)$$

where $W_B$ is effective monthly precipitation volume, P is precipitation volume, and $E_T$ is total evapotranspiration volume for the catchment, all expressed in mm. Using both GRACE and groundwater storage fluctuation datasets (see sections 2.3 and 2.4), accumulated net precipitation $W_{BI}$ was calculated as:

$$W_{BI} = \sum_n^1 W_B \qquad (2)$$

where $W_{BI}$ is net precipitation in month n. For any given month throughout the study period, $W_{BI}$ can be directly compared against GRACE and groundwater storage fluctuation data (mean long-term annual precipitation in the catchment is 270 mm).

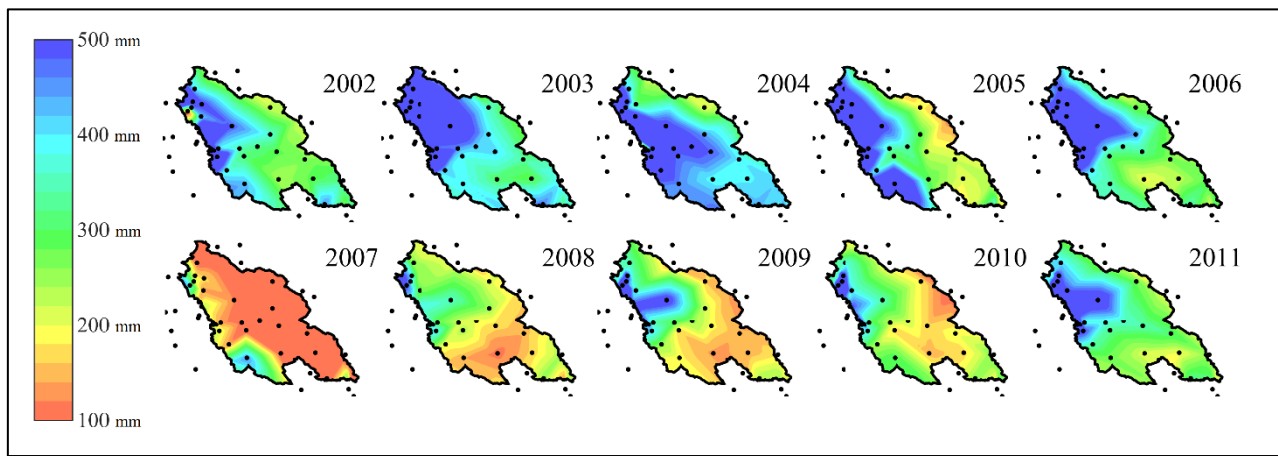

**Figure 2:** *Distribution of the 22 precipitation stations inside and around the Bakhtegan catchment, and the interpolated annual precipitation rate (in mm) during the period January 2002 to December 2011.*



## 2.3 Estimating Groundwater Storage Changes

Data on monthly groundwater levels in the Bakhtegan catchment are available from the 448 observation wells spread throughout the catchment (see Figure 1). As GRACE data show the monthly variation in total water mass in the Bakhtegan catchment, the specific monthly groundwater volume variation ($\Delta$s) in the total catchment area was calculated as:

$$\Delta S = \sum \frac{S_I \cdot A_I}{A_T} \tag{3}$$

where $S_I$ is the groundwater level variation in a given aquifer I, $A_I$ is the area of aquifer I, and $A_T$ is the total catchment area. Here, $\Delta_S$ is expressed in mm. For any given period, the accumulated groundwater volume variation $\Delta_{SI}$ is equal to:

$$\Delta_{SI} = \sum \Delta_S \tag{4}$$

In this case, $\Delta_{SI}$ shows the variation in groundwater storage volume. The groundwater extracted in the catchment can be assumed to be used mostly by agriculture, and thus directly related to soil moisture. Therefore $\Delta_{SI}$ can be compared with the water mass volume fluctuation derived from GRACE data.

## 2.4 GRACE-Derived Water Storage Analysis

We used 117 monthly GRACE datasets from April 2002 to December 2011 to examine the total water mass variation in the Bakhtegan catchment. The data consist of monthly snapshots, which when analyzed reveal monthly anomalies in total water storage. The GRACE data show mass change, or equivalent water height, in this study expressed as monthly variation ($W_{MI}$) in mm.

The GRACE data were provided by the German Research Center for Geosciences (GFZ). In order to identify water storage changes from the spherical harmonic coefficients provided by GFZ, the following procedure was followed: First, we replaced C20 according to instructions outlined in GRACE Technical note 7 (Cheng et al., 2013). We then added Degree 1 according to the estimation by Swenson et al. (2008) to correct for geocenter motion. The Degree 1 Love number ($K_1 = 0.021$) was taken into

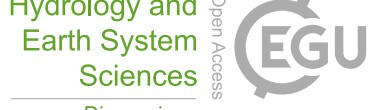



account. We then carefully removed the primary tidal aliasing error of the tidal constituents S2, S1, and P1, and the secondary tidal aliasing error of M2, O2, O1, and Q1 from the spherical harmonics (Tourian, 2013). Next, we removed the Glacial Isostatic Adjustment (GIA) according to the model provided by Wahr and Zhong (2013) and filtered the coefficients by a Gaussian filter (450 km radius) and a destriping filter, as proposed by Swenson and Wahr (2006). Finally, in order to account for leakage, we applied the so-called data-driven method developed by Vishwakarma et al. (2017). This method is able to restore the signal lost due to filtering, irrespective of the catchment size.

The groundwater component can be isolated from the total GRACE-derived data. According to Rodell and Famiglietti (2002), this can be achieved by subtracting the snow, surface water, and soil moisture monthly fluctuation volume from the total water mass:

$$GW_{MI} = W_{MI} - SW - SM \qquad (5)$$

where $GW_{MI}$ is the groundwater level derived from the GRACE data, $W_{MI}$ is the total water mass, SW is the surface water storage, and SM is the soil moisture. The change in surface water storage was calculated from flow data available for the Bakhtegan and Tashk lakes, which are the only surface water bodies in the catchment. Detailed daily flow volume data were also taken from Water Resources Atlas Report (2011). Soil moisture values were computed from the GLDAS datasets available from EARTHDATA (Rodell and Kato, 2007).

The Bakhtegan catchment area of 31 511 km$^2$ is much less than the effective limitation of GRACE data area, which is about 200 000 km$^2$ (Longuevergne et al., 2010). As GRACE data alone are not sufficient for analyzing catchment water mass fluctuations (Tourian et al., 2015), the data were compared against calculated net precipitation ($W_{BI}$) and groundwater volume variation ($\Delta_{SI}$). This was done by calculating two indices, $K_A$ and $K_B$:

$$K_A = \frac{W_{MI}}{W_{BI}} \qquad (6)$$

$$K_B = \frac{\Delta_{SI}}{W_{BI}} \qquad (7)$$



where $K_A$ shows the relationship between the GRACE data and net precipitation variation, i.e., it shows how both datasets are changing simultaneously within the study period, and $K_B$ shows the corresponding relationship between the GRACE data and groundwater volume variation.

## 3. Results

### 3.1 Comparing GRACE-Derived Data ($W_{MI}$) and Net Precipitation ($W_{BI}$)

The GRACE-derived data showed that total water mass storage ($W_{MI}$) in the region decreased by 76 mm from April 2002 to December 2011 (Figure 3A). This equated to an annual loss of approximately 7.6 mm and a total of 2.4 km$^3$ for the study period. The water storage volume showed a significant decreasing trend, especially after 2007, which was due to a regional drought in that year.

Monthly net precipitation calculated from the GRACE data showed a direct response to monthly sum of precipitation and evapotranspiration. With the decrease in annual precipitation, monthly net precipitation showed a negative trend during the study period. The monthly precipitation average data (Figure 3B) revealed a 25% decrease in rainfall volume for the year 2007. Mean annual rainfall in the catchment decreased from 307 mm in the period 2002-2006 to 230 mm in 2007. In the following four years (2008-2011), mean annual precipitation was 180 mm, which was significantly lower than the long-term annual mean of 270 mm. Evapotranspiration rates decreased by 15% in the period after 2007, and came relatively close to the annual average of 480 mm (Figure 3C). Owing to these changes in accumulated precipitation and evapotranspiration data, net precipitation ($W_{BI}$) showed a major decrease after 2007 (Figure 3D). The net precipitation was mostly constant in the early study period (2002-2006), while the evapotranspiration rate was on average 150 mm higher than precipitation in the post-drought period (2008-2011), leading to the trend of a loss in net precipitation ($W_{BI}$). This led for example to a decrease in the combined area of the Bakhtegan and Tashk lakes from 640 km$^2$ in 2002 to 220 km$^2$ in 2011, with the lowest recorded area of 71 km$^2$ in 2009.



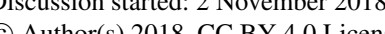

**Figure 3:** *Changes in [A] GRACE-derived water mass ($W_{MI}$), [B] total annual and monthly net precipitation (where the vertical line indicates mean long-term precipitation (270 mm) in the catchment), [C] evapotranspiration, and [D] climate water balance ($W_{BI}$) during the study period (January 2002-December 2011). The value of zero on the y-axis in A and D indicates no change in water mass.*

## 3.2 Comparing Observed Groundwater Level ($\Delta_{SI}$) and Net Precipitation ($W_{BI}$)

On average, the groundwater level decreased by 8 mm per month during the study period, although after the drought in 2007 the average decrease was 13 mm per month. The lowest annual groundwater levels occur at the end of the dry season, between September and October, while the highest levels occur



between April and May, due to precipitation and snow melt (Figure 4A). After the 25% decrease in precipitation in 2007, surface water scarcity increased the demand for groundwater for agriculture in the Bakhtegan catchment. This increase in demand increased the groundwater extraction rate, lowering the groundwater level. The accumulated groundwater volume variation ($\Delta_{SI}$) showed an increasing negative trend after 2007 (Figure 4D). In the same period the evapotranspiration rates remained constant, due to increased groundwater use for irrigation. The highest evapotranspiration rate, 120 mm, was recorded in summer 2007 (Figure 3C). Even though net precipitation ($W_{BI}$) still showed positive values in some periods after the drought, the GRACE data showed a negative water mass balance (Figure 4B). This loss of water mass in the catchment was directly related to both excessive groundwater extraction and increased evapotranspiration rate.

Monthly net precipitation and groundwater level fluctuation based on GRACE data showed good agreement. This was particularly evident for January and December 2004, when the highest positive net precipitation occurred (Figure 4C). Overall, during the study period the groundwater level fell by 905 mm, which indicated a significant water volume loss for the Bakhtegan catchment. However, the exact volume lost is difficult to calculate owing to the geology of the aquifers, which may share a boundary with other aquifers in surrounding catchments.





**Figure 4:** *[A] Measured monthly variation in groundwater level ($\Delta_S$), [B] variation in groundwater level ($\Delta_S$) compared with GRACE-derived water mass variation ($W_{MI}$), [C] variation in groundwater level ($\Delta_S$) compared with net monthly precipitation ($W_B$), and [D] accumulated groundwater height variation ($\Delta_{SI}$) compared with accumulated net precipitation ($W_{BI}$) during the study period (January 2002-December 2011). The value of zero on the y-axis in A, B, and C indicates no change in water mass.*

### 3.3 Comparing GRACE-Derived Groundwater Level (GW$_{MI}$) and Measured Groundwater Level ($\Delta_{SI}$)

The variation in groundwater levels in the catchments was assessed using both GRACE-derived groundwater data (GW$_{MI}$) and measured groundwater levels ($\Delta_{SI}$). The results showed a good fit



between monthly net precipitation ($W_B$), monthly measured groundwater level ($\Delta_S$), and monthly water mass change ($W_{MI}$) (Figures 4B and 4C). The $GW_{MI}$ and $\Delta_{SI}$ results showed a proportional fit, but were of different orders of magnitude. The groundwater mass storage change calculated using Equation 5 is compared with the in situ measured mass storage variation in Figure 5C, while Figures 5A and 5B show the variation in surface water and soil moisture storage, respectively. The soil moisture trend was -1 $\pm$ 1.5 mm annually. After the drought in 2007, the Bakhtegan and Tashk lake volume reached a minimum area as the inflow rate fell to zero. The water from the Kor River did not reach the lakes, as it was all used for irrigation. It was thus considered part of the soil moisture water mass in this study.





**Figure 5:** *[A] Change in measured monthly surface water storage, [B] monthly soil moisture storage change calculated from GLDAS data, [C] measured groundwater level variation ($\Delta_S$) compared with groundwater level derived from GRACE data ($GW_{BI}$), and [D] water mass ($K_A$) and groundwater level ($K_B$) fluctuation with respect to monthly net precipitation change in the Bakhtegan catchment during the study period (January 2002-December 2011). The value of zero on the y-axis in A, B, and C indicates no change in water mass.*

The two indices $K_A$ and $K_B$, which show the relationship between $W_M$ and $\Delta_S$ by comparing them to the monthly net precipitation ($W_B$) (Equations 6 and 7), are shown in Figure 5D. $K_A$ shows how the water mass in the catchment is directly related to net monthly precipitation, where the closer the value of $K_A$ is to zero, the less these two variables are directly related on monthly basis. The $K_A$ results showed that the water mass fluctuation in the catchment could not predicted from monthly precipitation and



evapotranspiration rates, as they comprised small volumes compared with the total water mass in the catchment. The water mass changes on catchment scale were more evident in annual periods. $K_B$ shows the relationship between net monthly precipitation and groundwater level variation. Unlike $K_A$, $K_B$ showed that the groundwater level variations before the year 2007 were directly proportional to the

precipitation and evapotranspiration rates. The aquifer system in the Bakhtegan catchment consists of shallow alluvial aquifers with a depth range between 30 and 50 m (Rasoulzadeh and Mosavi, 2007). Precipitation directly affects these shallow aquifers, which are usually smaller in volume than deeper confined aquifers. The results showed that groundwater level decrease in different aquifers of the Bakhtegan catchment ranged between 5 and 30 m during the study period, with an overall average

decrease of 10 m for the catchment (Figure 6). The eastern and northern plains in the catchment suffered the greatest decrease in groundwater level. Due to the increase in groundwater extraction after 2007, the most intense groundwater level decrease occurred between 2007 and 2011. The $K_B$ results show that, in the same period, that effect of net monthly precipitation on groundwater levels decreased. The groundwater level in this period might have reached the confined aquifers beneath the alluvial

aquifers, where the direct effect of precipitation on the recharge rate is lower. However, as these confined aquifers are considered to have a much higher volume than the shallow aquifers, any change in their volume has a direct change on the total water mass in the Bakhtegan aquifer.

The groundwater stored in confined aquifers is subjected to compression from the overlying soil particles and water. Thus the groundwater level loss recorded in this study may be related to pressure

stabilization in the confined aquifers. If extraction wells have started to reach into water in the deeper confined aquifers, the groundwater elevation data reflect the change in pressure, not the emptying of soil pores. The storage coefficient in confined aquifers is considerably smaller than in unconfined, alluvial aquifers, as only pressure, and not pore water, is released. In theory, this can also explain the drop in groundwater level after a recorded increase in water level in the year 2007 (Figure 4D).

Increasing the depth of wells to provide water resources after the drought in 2007 may have caused fluctuations due to pressure changes on reaching the confined aquifers. $K_B$ also shows the change in fluctuation ratio when comparing unconfined and confined aquifer systems. The groundwater level was



highly proportional to the precipitation rate before 2007, while the effect of rainfall volume decreased after 2007 (Figure 5D).



**Figure 6:** *Total decrease (m) in groundwater level in different alluvial aquifers of the Bakhtegan catchment between January 2002 and December 2011.*



## 4. Discussion

Aquifers are essential and critical water sources in semi-arid regions. With increasing area of irrigated agriculture, groundwater levels have declined considerably. In this study, we showed that measuring the decline in water resources independently at the aquifer scale with GRACE data provides a unique way to obtain data on aquifer water use. These data, when applied in integrated local water management plans, can help achieve development goals for semi-arid rural areas, which include providing clean water for both domestic and agricultural use. In particular, GRACE data can help draw general conclusions about aquifer conditions when site observations are lacking.

At regional scale, such as in the Bakhtegan catchment, water stress is a major risk. Population growth, urbanization, and increases in agricultural and industrial activities have all led to an increase in demand for fresh water. During the drought year of 2007, there was an increase in irrigation demand and evapotranspiration rates also increased, with both factors contributing to increasing extraction rates from Bakhtegan catchment aquifers. Based on the GRACE data, the estimated total water loss was nearly 2.4 km$^3$ (7.6 mm annually), a severe water loss for a semi-arid area such as the Bakhtegan catchment, which is already facing water scarcity. The results showed that groundwater depletion in the catchment is the major water volume loss causing a negative trend in water mass. As annual net precipitation became negative during the latter part of the study period, groundwater extraction may have reached confined fossil aquifers, affecting the sustainability of these resources. This water depletion directly affected the agricultural sector, as the cultivated area in Iran decreased by 2 million hectares between 2008 and 2012 (AQUASTAT, 2015). This decrease in cultivated area resulted in negative economic revenue in rural areas and increased the urbanization rate.

With sufficiently long series of GRACE data now available, many regional-scale studies for validation have been conducted. Comparisons between modeled outputs and measured data have revealed acceptable agreement between GRACE-derived and total water storage variation, e.g., Rodell et al. (2007) found good agreement between monitored groundwater levels and GRACE-derived values in the Mississippi River basin. A study using the same methodology confirmed groundwater depletion in the





Rajasthan plain in north-western India (Rodell et al., 2009). Seasonal correlation factors ranging between 0.8 and 0.9 have been found when comparing GRACE data with site measurements in aquifers in Illinois (Yeh et al., 2006) and Oklahoma (Swenson et al., 2008). GRACE data also provide an alternative solution for studies of transboundary aquifers located in different countries, e.g., Voss et al.

(2013) observed groundwater depletion in the Tigris-Euphrates basin, which is shared between Turkey, Syria, and Iraq.

Our study highlights the importance of GRACE satellite data for hydrological analyses to support environmental and water management decisions. Although site-based observations are still seen as the most reliable information source, this study demonstrated that GRACE-derived data were able to

confirm the water volume loss determined in the catchment during the study period. The monthly net precipitation trend showed a good fit with the GRACE data, and had direct effects on catchment water mass. Although there were some discrepancies in monthly groundwater depth, the GRACE data showed a loss in water mass volume similar to that based on the site observations of groundwater level. The discrepancies can be due to the small study area (31 511 km$^2$) (Longuevergne et al., 2010), hydraulic

interaction with neighboring catchments, or changes in soil porosity, which can delay the water mass change in the catchment. The variation in order of magnitude between the results obtained with the two methods might also be related to aquifer geology and geometry. However, some general conclusions can be drawn from the results.

There is a lack of data in developing countries, possibly due to lack of funding, technology, and

logistics, and the GRACE satellite can provide important hydrological data for these countries. Such data are essential for any serious attempts in establishing catchment-level or national integrated water resources management plans. In semi-arid to arid climate zones, groundwater is the only effective water source during long dry seasons and drought years. This is basically due to its relatively higher availability, cheaper exploitation methods, and good quality for direct use and irrigation. In developing

countries, where most societies are rural, groundwater is a strategic resource, as it is for farming communities in Bakhtegan catchment after a prolonged drought. Future advances in hydrological remote sensing and modeling are likely to provide the opportunity for creating a more accurate and



wider image of the availability of renewable water resources. This can increase the effectiveness and transparency of water management in regions with water conflicts and also for transboundary basins.

## 5. Conclusions

In this study, we performed a direct comparison between GRACE-derived data, measured groundwater levels, and calculated net precipitation in a semi-arid zone. The results showed good agreement between observed groundwater levels and calculated monthly net precipitation, and between GRACE-derived data and groundwater level fluctuations. During the study period, the groundwater level dropped by an average of 10 m in the Bakhtegan catchment and in some plains areas a decrease of 30 m was recorded. This was mainly due an increase in groundwater exploitation, particularly for irrigation in agriculture, the main water-consuming activity in the catchment, especially after a drought in 2007. Deeper wells were dug at that time, reaching into confined aquifers. Thus the increased groundwater exploitation used non-renewable resources, a critical issue for water resources management in semi-arid zones. Based on the results in this and previous studies, GRACE data allow general conclusions to be drawn regarding catchment water mass change. This provides an opportunity to study areas for which in situ data are scarce or unavailable. As groundwater is considered the most valuable resource for rural communities in semi-arid and arid zones, these data are important for integrated water resources management for these communities.

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
