# Peer review of "Monitoring Groundwater Storage Depletion Using Gravity Recovery and Climate Experiment (GRACE) Data in the Semi-Arid Catchments"

_Hydrology and Earth System Sciences, 2018_

## Short Comment (SC1) · 11 Dec 2018

In this study, data from the GRACE satellite program was used to monitor groundwater depletion problems in a semi-arid environment. The study attempted to establish a correlation between GRACE data and in situ measurements to create a method to assess and monitor other catchments suffering from extreme groundwater depletion. The main points of this paper include; judging the efficiency of GRACE data in catchments smaller than the recommended limit of 200 000 km2, as well as comparing the GRACE-derived data and in situ measurements groundwater depletion estimates. The study focused on a catchment with an area less than 16% of the proposed GRACE

area limit (200 000 km2) with an aquifer area of less than 5%, despite recently published work showing that data for catchments below 63,000 km2 becomes too noisy for interpretation (Vishwakarma, Devaraju, & Sneeuw, 2018). To address this limitation, two indices were calculated, comparing GRACE water mass and in situ groundwater volume variation to calculated net precipitation. Results of the indices showed that GRACE data could not be related to precipitation while in situ measurements were directly proportional to precipitation and evapotranspiration rates. Due to the relative commonality of papers using GRACE data to examine groundwater depletion, and its focus on a catchment smaller than the effective limitation of GRACE data this paper does not stand out nor make a significant impact.

Major Comments:

1) The section "Estimating Groundwater Storage Changes" is unclear as to whether $\Delta S$ is calculated from observation wells or GRACE data. The opening paragraph mentions both methods, thus confusing the reader.

2) The main point of this paper was to assess the catchment using GRACE-derived water mass data (WMI) against in situ monthly measurements. To analyze this, the methods section describes isolating the groundwater component (GWMI) given in Equation 5. Yet minimal explanation of this is made in the results section and little is discussed other than the fact that GWMI and $\Delta ĂňSI$ results show a proportional fit. What is the average monthly variation of GWMI?

3) Calculated groundwater depletion estimated from $\Delta ĂňSI$ was determined to be 905 mm; GRACE data showed a 76 mm decline. However, results showed a groundwater level average depletion of 10 m in the catchment aquifers. Where does this value come from? This seems to disagree with the suggestion that the GRACE-derived data and groundwater level fluctuations showed good agreement, as presented in the conclusion section.

4) It is confusing throughout the paper to what the KB indices represent. It is presented

as representing GRACE data in some cases, and representing in situ measurements in others, despite being defined as KB = ΔSI / WBI in Equation 7. Examples:

- P7, L19: "GRACE data alone are not sufficient for analyzing catchment water mass fluctuations (Tourian et al., 2015), the data were compared against calculated net precipitation (WBI) and groundwater volume variation (ΔSI)"

- P8, L2: "KB shows the corresponding relationship between the GRACE data and groundwater volume variation."

- Figure 5: "water mass (KA) and groundwater level (KB) fluctuation with respect to monthly net precipitation change in the Bakhtegan catchment"

- P14, L2: "KB shows the relationship between net monthly precipitation and groundwater level variation"

Specific comments:

5) P1, L17: Remove comma in 200,000 km2 to remain consistent with formatting in the remainder of the paper

6) P4, L2: The paper mentions that GWMI and ΔÂňSI results show a proportional fit. However, there is no graph of these results available in the report.

7) P6, L22: add the word values after C20 to improve the clarity of the statement.

8) P7, L6: Remove "so-called" from the sentence, or substitute with "proposed"

9) P7, L15: was taken, instead of "were also taken"

10) P7, L18: Grammar. As GRACE data alone are is not sufficient for analyzing catchment water mass fluctuations (Tourian et al., 2015), the data were is compared against calculated net precipitation (WBI) and groundwater volume variation (ΔSI).

11) P8, L2: KB is stated as the relationship between GRACE data and groundwater volume variation. However, it is defined as ΔÂňSI/WBI (Accumulated groundwater

volume variation/Net precipitation in month n) which are both in situ measurements. Which is correct?

12) P8, L11: Consider adding "the" before "monthly sum of"

13) P10, L1: Snowmelt is one word

14) P10, L4: Consider replacing "increasing" with "increasingly"

15) P13, L11: include "a" in between on and monthly to improve sentence flow

16) P14, L2: water mass changes on "the" catchment scale...

17) P14, L8: results showed that "the" groundwater level decrease

18) P16, L1: With "an" increasing area of irrigated

19) P:17, L23: Remove "basically" from the sentence

20) Figure 1: Increase the quality of the inserts, specifically insert A.

21) Figure 5C: Vertical axis variables do not match those listed in the figure description. GRACE data (GWBI) should be GRACE data (GWMI)

22) Figure 6: What do the black lines surrounding the aquifers represent?

Overall, this paper is weak, and it is not clear how the work goes beyond the status quo in the GRACE literature. The generally confusing formatting and explanations of findings throughout make publication in HESS at this time likely not possible.

Before this paper is ready for publication, an effort should be made to clarify the methods and results to improve readers comprehension, as well as emphasizing the need for another paper investigating groundwater depletion using GRACE data and justifying the viability of interpreting GRACE data on a catchment smaller than the recommended area.

Reference

Vishwakarma BD, Devaraju B, Sneeuw N. What Is the Spatial Resolution of grace Satellite Products for Hydrology? Remote Sensing. 2018; 10(6):852.

---

## Referee Comment (RC1) · Karimi (Referee) · 2 Jan 2019

The paper is interesting in the context of serious water issue in the region, especially regarding the groundwater. The country is suffering critically from nonavailability of a correct mapping of the extent of problem, which is the basis of correct management setup. This paper suggests an independent approach to data collection and evaluation, which is very important. However, I suggest the publication of this paper after the authors address the following questions more clearly in the paper.

• How did the smaller study area affect the GRACE derived results

• How does the geology of the catchment interrupt with the water mass variation

• How does the water mass change in the catchment indicates the groundwater recharge and depletion

• What are other factors interrupting the water mass variation, other than discussed in the paper

---

## Referee Comment (RC2) · Forootan (Referee) · 28 Jan 2019

I read the paper by Abou Zaki et al. ('Monitoring Groundwater Storage Depletion Using Gravity Recovery and Climate Experiment (GRACE) Data in the Semi-Arid Catchments') with an interest. The authors try to apply (GRACE) data to determine water mass changes in a catchment located in the semi-arid part of Iran. Through this specific assessment, the authors claim that their approach is suitable for all semi.arid catchments, which I cannot approve their claim based on the current investigation. For this, and many technical issues, I recommend a reject outright decision. Major comments: Title: in the Semi-Arid Catchments –> the study only considers one catchment

in a particular region. How come does the title claim it plural!? The introduction is too short and ignore even the GRACE related studies of the region.

Various studies e.g., Forootan et al 2014 and 2017 provide a complete analysis of water storage and water fluxes in the area. Joodaki et al 2014 assessed GRACE against well observations. Forootan, E., Safari, A., Mostafaie, A. et al. (2017) Large-Scale Total Water Storage and Water Flux Changes over the Arid and Semiarid Parts of the Middle East from GRACE and Reanalysis ProductsSurv Geophys (2017) 38: 591. https://doi.org/10.1007/s10712-016-9403-1 Forootan E, Rietbroek R, Kusche J, Sharifi MA, Awange JL, Schmidt M, Famiglietti J (2014) Separation of large scale water storage patterns over Iran using GRACE, altimetry and hydrological data. Remote Sens Environ 140:580–595. doi:10.1016/j.rse.2013.09.025 Joodaki G, Wahr J, Swenson S (2014) Estimating the human contribution to groundwater depletion in the Middle East, from GRACE data, land surface models, and well observations. Water Resour Res 50:2679–2692. doi:10.1002/2013WR014633 These studies have already assessed the GRACE data against wells. What is the innovation of this study against the already published works?

The values that are reported as the decline of groundwater do not match?

---

## Referee Comment (RC3) · Anonymous Referee #3 · 30 Jan 2019

I have reviewed the manuscript by Abou Zaki et al. which evaluates groundwater storage changes in the Bakhtegan catchment in south-western Iran. I find their results interesting; however, I don't feel that the manuscript is appropriate for publication in its current form. I have several primary concerns. First of all, the novelty of this work is not clear. As other reviewers have pointed out, there are a myriad of studies using GRACE to evaluate groundwater storage changes and compare with well data. In my opinion the authors do not do a sufficient job putting this work in the context of that body of research and highlighting what is novel about this study. Second, the methods are unclear and there are inconsistencies in the descriptions and terminology that make it very difficult to determine exactly what is being calculated and compared. Finally,

there is no uncertainty analysis and the authors have not demonstrated that the trends they report are statistically significant. I have outlined more detailed comments below. However, given these significant deficiencies I recommend rejecting the manuscript.

Title: This is really a study of one basin not a study of semi-arid basins in general and the title should be revised to reflect this.

Introduction: There is some reference to previous GRACE work in the introduction, but additional discussion is needed to better explain how this specific location adds information to the comparisons that have already been done.

Methods: There are many points in the methodology that are inaccurate or are not clear. I have listed some examples below:

- In equation 3 how is porosity accounted for?

- There is no detail on how the well data was combined to get the groundwater storage estimates and what hydrogeologic properties were used.

-It is also unclear how the ET rates were calculated. What time period were the 'average ET rates' averaged over? And how was land use data used to adjust these values for agricultural land and forests? A figure showing the ET with some uncertainty bounds on ET would be helpful.

-In equation 5 why do the GW and total Water components have the MI subscript but not the other two?

-Equation 5 talks about total storage but to be consistent with GRACE these should be changes in storage correct? I don't see anywhere here where the total groundwater storage is calculated.

-More details on the surface storage calculations as well as soil moisture would also be helpful. For example, where are the stream gauges located? How exactly were storage changes calculated taking into account the Water Atlas report?

-It looks like Equation 2 assumes that all of the net precipitation goes to recharging groundwater. What about runoff? How is this accounted for?

-Page 8 Line 1: Please specify exactly what 'the GRACE data' you are referring to here.

-Page 8 Line3: Contrary to the text, It looks like KB is actually the relationship between groundwater volume changes and precipitation not GRACE? Is this correct?

-Page 8 Line 11: How was monthly net precipitation calculated from GRACE? This does not make sense to me.

Terminology:The terms groundwater storage, and groundwater levels are used interchangeably in the manuscript. I think the authors should be more careful in the definition and use of these terms. This especially needs clarification because the storage changes are expressed in length units which can give the impression that they reflect groundwater depth changes, when they do not.

Additional Comments:

-Page 10 Lines 13-16: The authors note it is difficult to calculate volumetric losses from the groundwater measurements. However, this is one of the key goals of the study and is necessary for a meaningful comparison to GRACE. As noted above, I would like additional details on how groundwater storage was calculated and what the uncertainties in this calculation are.

-Page 11 Line 12: What constitutes a 'good fit'? Can you be more precise in how this was quantified?

-Page 14 Lines 15-17: This sentence doesn't make sense to me, please revise.

-Page 14 paragraph starting on lines 18: This discussion is a bit hard to follow. I think it would fit better in the methods section where the details on how groundwater storage changes are calculated should be included. It's not clear from this discussion how the

confined and unconfined units were treated in the delta S calculation.

-Discussion Page 15 Lines 3-5: it is unclear how the combination of these datasets provides unique information on water use. Mainly what has been presented here is a comparison between methods and it's not clear that the difference between these two reflects anything other than uncertainty in all of the water balance components. Please clarify what you mean by this.

-Page 16 lines 14-15: I disagree that 7.6 mm is a 'severe water loss' even for a semi-arid area. Can you provide some justification for this classification?

-Page 16 line 18: The analysis presented here does not do anything to prove that 'confined fossil aquifers' were reached so this should not be included in the discussion.

-The discussion section is really broad and connects out to too many things not covered in the results. I recommend refocusing the discussion around the findings of the paper as well as the uncertainties and assumptions that were made and the potential implications of these limitations.

-The authors talk about water level drops of 10 -30 m over the 10 year study period which but then have annual trends of 7.6mm. In order to rectify these measurements we need a better understanding of the physical properties of the aquifer. Without knowing this it is difficult to understand how reasonable the trends in volume are.

---

## Author Comment (AC1) · 7 Feb 2019

The aim of this study is to assess the efficiency of the (GRACE) derived data in catchments smaller than the recommended limitation of 200,000 squared kilometers suggested by (Longuevergne et al., 2010). This approach was stated clearly in the abstract (Page 1: Line 16-17), and discussed later in the discussion section (Page 17: Line 7-18). When approached on smaller scale, GRACE data tend to show more uncertainty. For a better understanding, Bekhtegan catchment is chosen as a study site, as detailed data is available from 448 groundwater observation wells, and 22 climatic monitoring station. The water mass balance of the catchment is calculated from this collected

data, which includes the daily precipitation, evapotranspiration and soil moisture. This water mass is compared with GRACE derived data (Figure 4B and 4C). Also the two constants (KA) and (KB) (Page 7: equations 6 and 7) (Figure 5D) show the uncertainty when comparing the (GRACE) data with both the water mass and the groundwater volume variation. The study also suggests that the uncertainty can also be related to the catchment's aquifers type, as the aquifers in the study area are considered to be shallow (Page 14: Line 5-10) and (Page 14 Line 20-25).

We agree with the referee statement that a lot of studies have been done to estimate groundwater fluctuation using (GRACE) data. Most studies are older than the studies suggested by the reviewer: {Various studies e.g., Forootan et al 2014 and 2017 provide a complete analysis of water storage and water fluxes in the area}. Voss et al. (2013), Tourian et al. (2015) as Middle East cases, and other studies from different study regions mentioned in (Page 16 Line 22 till Page 17 Line 6). In the introduction (Page 2 Line 11-19) also mentions, as much as the introduction section will allow, the (GRACE) data usage and previous studies. But what all these previous studies have in common, is the large scale study area. In our study, smaller scale helps noticing the direct effects of local hydrologic phenomena like droughts (Page 8: line 6-24) on the groundwater level and its occurrence in (GRACE) derived data. The study discusses the efficiency of (GRACE) data as a tool that can be used for water management on local level (Page 17 Line 19 till Page 18 Line 2). From what mentioned, we believe this study gives innovation against already published studies.

We agree on the (major comment) of the referee regarding the title, even that the approach can be used in any semi-arid catchment. If accepted by the editor for the next level of revision, we will change the title to agree with the reviewer comment.

---

## Author Comment (AC2) · 13 Feb 2019

We very much appreciate the review of our anonymous referee. The insightful and comprehensive comments helped us to make numerous changes in the way how our data and methodology are interpreted, presented and discussed. After making these changes, we will appreciate if the referee would suggest publishing of the article.

The aim of this study is to assess the efficiency of the (GRACE) derived data in catchments smaller than the recommended limitation of 200,000 km2 suggested by (Longuevergne et al., 2010). This approach was stated clearly in the abstract (Page 1: Line 16-17), and discussed later in the discussion section (Page 17: Line 7- 18). When

approached on smaller scale, GRACE data tend to show more uncertainty. For a better understanding, Bekhtegan catchment is chosen as a study site, as detailed data is available from 448 groundwater observation wells, and 22 climatic monitoring station. The water mass balance of the catchment is calculated from this, which includes the daily precipitation, evapotranspiration and soil moisture. This water mass is compared with GRACE derived data (Figure 4B and 4C). Also the two constants (KA) and (KB) (Page 7: equations 6 and 7) (Figure 5D) show the uncertainty when comparing the (GRACE) data with both the water mass and the groundwater volume variation. The study also suggests that the uncertainty can also be related to the catchment's aquifers type, as the aquifers in the study area are considered to be shallow (Page 14: Line 5-10) and (Page 14 Line 20-25). All the previous (GRACE) studies have in common the large scale study area. In our study, smaller scale helps noticing the direct effects of local hydrologic phenomena like droughts (Page 8: line 6-24) on the groundwater level and its occurrence in (GRACE) derived data. The study discusses the efficiency of (GRACE) data as a tool that can be used for water management on local level (Page 17 Line 19 till Page 18 Line 2). From what mentioned, we believe this study gives innovation against already published studies. Below are our point by point answers for the referee comments:

* Title: This is really a study of one basin not a study of semi-arid basins in general and the title should be revised to reflect this:

We agree on this comment, and the title can be changed to (Monitoring Groundwater Storage Depletion Using Gravity Recovery and Climate Experiment (GRACE) Data in Bakhtegan Catchment).

* Introduction: There is some reference to previous GRACE work in the introduction, but additional discussion is needed to better explain how this specific location adds information to the comparisons that have already been done:

The introduction mentions quickly some previous studies, and more studies are mentioned in the discussion section. In the revised version, a paragraph is added to compare our finding with previous studies finding. This can lead to general conclusions about the effects of the specific location on the results.

* Methods: There are many points in the methodology that are inaccurate or are not clear. I have listed some examples below: - In equation 3 how is porosity accounted for:

In equation 3, ($\Delta$S) is expressed in (mm) and shows the actual variation of the groundwater levels. Here the porosity doesn't affect the calculation, as the groundwater level here is based on actual site measurements. ($\Delta$S) is the difference in measurements between two consecutive readings. Actually porosity rate in used in equation (2), (WBI), which is the net precipitation volume. According the Atlas report, the precocity volume fraction is considered 0.1 (Reference will be added in revised version)

* There is no detail on how the well data was combined to get the groundwater storage estimates and what hydrogeologic properties were used:

Section (2.3) (Page 6) describes how the groundwater level variation in the catchment was calculated. Equation (3) shows the conversion of the monthly reading of the observation wells (in meters) to a variation in (mm), on the sub-catchment level. In Equation (4) the variation in the whole catchment is calculated. Expanding the explanation of the methodology in the revised version will give a better understanding of the last two comments.

* It is also unclear how the ET rates were calculated. What time period were the 'average ET rates' averaged over? And how was land use data used to adjust these values for agricultural land and forests? A figure showing the ET with some uncertainty bounds on ET would be helpful:

(Line 7 Page 4 till Line 1 Pages 5) states that the land use data and the evapotranspiration data was estimated from images and data available on (USGS Modis) and

(Climengine.appspot.com). The evapotranspiration data is hourly data referred to different land use zones in the catchment. The data covers the same study period from April 2002 till December 2011. We agree with the review that an equation must be added, before equation (1), regarding the calculation of the average evapotranspiration in the catchment (mm). The average evapotranspiration rate is shown in (figure 3C).

* In equation 5 why do the GW and total Water components have the MI subscript but not the other two:

(MI) is added in this equation to separate (WMI) as the total water mass, from (WBI) as the net monthly effective precipitation volume mentioned in equation (2). (SW) and (SM) and monthly volume on surface water and soil moisture variation, and were mentioned in this equation only. (MI) is used to differentiate two similar constants.

* Equation 5 talks about total storage but to be consistent with GRACE these should be changes in storage correct? I don't see anywhere here where the total groundwater storage is calculated:

Please refer to (Page 7 – Line 8): The GRACE data shows the total water mass variation in the catchment. According to (Rodell and Famiglietti, 2002) to get the groundwater volume variation from GRACE, other water components masses must be eliminated. Here we eliminate the surface water volume, based on calculation of the Bakhtegan and Tashk lakes volume variation, and the soil moisture volume, which was computed from GLDAS datasets as mentioned.

* More details on the surface storage calculations as well as soil moisture would also be helpful. For example, where are the stream gauges located? How exactly were storage changes calculated taking into account the Water Atlas report:

We agree that this calculations must be shown in more details in the methodology section. Lake Bakhtegan and Tashk are the only sources of surface water in the catchement. The stream gauge are located in the lakes inlet. The calculation in the surface

water volume is calculated from the flow data recorded, subtracting the lake evapo-transpiration. The soil moisture data is the average of the soil moisture in the three measured soil layers. This data as mentioned is computed from GLADS database. This will be mentioned in the revised version.

* It looks like Equation 2 assumes that all of the net precipitation goes to recharging groundwater. What about runoff? How is this accounted for:

As mentioned before, the porosity volume factor considered for this calculation is 0.1 as recommended. This will be added with reference to the manuscript.

* Page 8 Line 1: Please specify exactly what 'the GRACE data' you are referring to here:

In page 7, equation (5) states that the GRACE data intends to mean the groundwater volume variation derived from GRACE data. This will be mentioned in the revised version

* Page 8 Line3: Contrary to the text, It looks like KB is actually the relationship between groundwater volume changes and precipitation not GRACE? Is this correct:

True, (KB) shows the relationship between the groundwater volume changes and the precipitation. This will be corrected in the revised version

* Page 8 Line 11: How was monthly net precipitation calculated from GRACE? This does not make sense to me:

This sentence (Page 8 line 11) must be corrected from (Monthly net precipitation calculated from GRACE) to (Monthly groundwater volume variation derived from GRACE)

* Terminology: The terms groundwater storage, and groundwater levels are used interchangeably in the manuscript. I think the authors should be more careful in the definition and use of these terms. This especially needs clarification because the storage changes are expressed in length units which can give the impression that they

reflect groundwater depth changes, when they do not:

Equation (3) and (4) (Page 6 Line 6 and 10) defines the groundwater storage variation derived from the in-situ data, while equation (5) (Page 7 Line 11) defines the groundwater storage derived from GRACE data. Groundwater level are mentioned a variable used for calculating equation (3 and 4) in (Page 6 Line 7). It will be clarified that the groundwater depth in (mm) is a unit of measuring the storage changes and not the depth changes.

* Page 10 Lines 13-16: The authors note it is difficult to calculate volumetric losses from the groundwater measurements. However, this is one of the key goals of the study and is necessary for a meaningful comparison to GRACE. As noted above, I would like additional details on how groundwater storage was calculated and what the uncertainties in this calculation are:

As answered before the methods of calculating groundwater storage from in-situ and GRACE data have been clarified. Here we intend to show that the difference in the volume variation from the two data sources, can be referred to aquifer geology and transboundary flow from surrounding catchments.

* Page 11 Line 12: What constitutes a 'good fit'? Can you be more precise in how this was quantified:

"Good Fit" was related to the results of (Figures 4B and 4C). In (Figure 4C) it can be noticed that ($\triangle$S) and (WB) have almost identical results. In the (Figure 4B), the both ($\triangle$S) and (WMI) show negative trend and decrease in the storage volume.

* Page 14 Lines 15-17: This sentence doesn't make sense to me, please revise:

This will be revised in the meaning of: The lower confined aquifers have higher volume compared to the upper alluvial aquifer. Any depth changes in these aquifers, will have a direct change on the water volume storage change.

* Page 14 paragraph starting on lines 18: This discussion is a bit hard to follow. I think

it would fit better in the methods section where the details on how groundwater storage changes are calculated should be included. It's not clear from this discussion how the confined and unconfined units were treated in the delta S calculation:

(Figure 4D) shows that post the drought period of the year 2007, there was an increase in the groundwater exploitation volume. This can be due to the increase in the groundwater usage to adjust the low precipitation volume. Also the drop in the groundwater level, as the paragraph starting on line 18 states, might also be related to pressure stabilization due to the increase of the well depth. This means that the level drop can be related to two factors, rather than just over exploitation rates. Confined and unconfined units are involved in the ($\triangle$S) calculation.

* Discussion Page 15 Lines 3-5: it is unclear how the combination of these datasets provides unique information on water use. Mainly what has been presented here is a comparison between methods and it's not clear that the difference between these two reflects anything other than uncertainty in all of the water balance components. Please clarify what you mean by this:

I think you meant (PAGE 16, Line 3-5). In the same paragraph (Line 7-8) we mentioned that GRACE data can help draw general conclusions about aquifer conditions. Here referring to (Figure 5C), GRACE showed a negative trend in the groundwater storage volume. Also later in the discussion, we mentioned that the area of the catchments directly can be related to uncertainty of the GRACE results, but still referring to the word (general), some conclusions can be drawn. This, and what stated in the first page of the authors response, is what mainly presented.

* Page 16 lines 14-15: I disagree that 7.6 mm is a 'severe water loss' even for a semiarid area. Can you provide some justification for this classification:

7.6 mm is the annual volume depth lost on the catchment level, which is equivalent to 2.4 km3 in the period of 10 years. Referring to (Figure 6 Page 15) that loss reached 30 meters in groundwater table level in some aquifers. This decrease, increased the wells

depth in many aquifers

* Page 16 line 18: The analysis presented here does not do anything to prove that 'confined fossil aquifers' were reached so this should not be included in the discussion:

As part of our discussion we used the word (have) to indicate that this might be a case. Moreover previously we mentioned that drop in the groundwater level might be due to pressure stabilization when reaching confined aquifers. Also as mentioned before we know from the study area that the well depth has increased

* The authors talk about water level drops of 10 -30 m over the 10 year study period which but then have annual trends of 7.6mm. In order to rectify these measurements we need a better understanding of the physical properties of the aquifer. Without knowing this it is difficult to understand how reasonable the trends in volume are:

The 7.6 mm annual volume depth lose is calculated considering the catchment as one aquifer. The more detailed sub-aquifer lose is presented in (Figure 6 page 15), where the properties of the aquifer play an important role in this level loss. Also and assumption discussed in the discussion section when describing the areas aquifers as shallow aquifers

---

## Author Comment (AC3) · 15 Feb 2019

We want to thank the reviewer for his time and effort commenting our manuscript. We are aware that GRACE data have been used to examine groundwater depletion and that many studies have been published before. Still this study relies also on in-situ data regarding groundwater level (448 observation wells) and 22 observation station measuring the precipitation, evapotranspiration and flow on hourly bases. All the previous (GRACE) studies have in common the large scale study area. In our study, smaller scale helps noticing the direct effects of local hydrologic phenomena like droughts (Page 8: line 6-24) on the groundwater level and its occurrence in (GRACE) derived

data. The study discusses the efficiency of (GRACE) data as a tool that can be used for water management on local level (Page 17 Line 19 till Page 18 Line 2). Below are our point by point answers for the reviewer comments:

* The section "Estimating Groundwater Storage Changes" is unclear as to whether $\Delta S$ is calculated from observation wells or GRACE data. The opening paragraph mentions both methods, thus confusing the reader:

In the same section, please refer to (Page 6 – Line 1): data on monthly groundwater levels in the Bakhtegan catchment are available from 448 observation well. Also refer to (Line 13): Therefor ($\Delta SI$) can be compared with water mass volume fluctuation derived from GRACE data. Both of these sentences show that ($\Delta S$) refers to the in-situ data collected. Still this can be made clearer in the reviewed version

* The main point of this paper was to assess the catchment using GRACE-derived water mass data (WMI) against in situ monthly measurements. To analyze this, the methods section describes isolating the groundwater component (GWMI) given in Equation 5. Yet minimal explanation of this is made in the results section and little is discussed other than the fact that GWMI and ($\Delta SI$) results show a proportional fit. What is the average monthly variation of GWMI?

(Page 7 – Line 11 – Equation 5): equation 5, referred to a paper published by Rodell and Famiglietti (2002), indicated that in order to compare the GRACE derived data with the in-situ data, all the water bodies in the catchment must be eliminated. As this is a semi-arid catchment, besides groundwater bodies, surface water from Bakhtegan and Tashl Lakes, and soil moisture are considered the only water bodies. This is regarding the methodology. Regarding the results, the (GWMI) monthly variation are equivalent to (WMI) values, as the surface water and soil moisture variation are negligible compared to the groundwater volume variation. Groundwater is the major water body in the catchment. This can be clarified more in the revised version.

* Calculated groundwater depletion estimated from ($\Delta SI$) was determined to be 905

mm; GRACE data showed a 76 mm decline. However, results showed a groundwater level average depletion of 10 m in the catchment aquifers. Where does this value come from? This seems to disagree with the suggestion that the GRACE-derived data and groundwater level fluctuations showed good agreement, as presented in the conclusion section:

An average of 10 meter of depletion is the actual groundwater level decrease rate in the period of 10 years. The 905 mm is the water volume (depth) lost in the same period. This two results show two different things, and we guess the reviewer mixed between volume depth, and groundwater levels. For that if you review (Figure 4D) you will find a good fit between the two. Comparing the GRACE derived data results and the in-situ data, can be found in the discussion part, explaining these results. This can be because of small catchment size or because of the pressure stabilization mentioned in (Page 14 Line 18 – Page 15 Line 2).

* It is confusing throughout the paper to what the KB indices represent. It is presented as representing GRACE data in some cases, and representing in situ measurements in others, despite being defined as KB = ΔSI / WBI in Equation 7. Examples: - P7, L19: "GRACE data alone are not sufficient for analyzing catchment water mass fluctuations (Tourian et al., 2015), the data were compared against calculated net precipitation (WBI) and groundwater volume variation (ΔSI)" - P8, L2: "KB shows the corresponding relationship between the GRACE data and groundwater volume variation." - Figure 5: "water mass (KA) and groundwater level (KB) fluctuation with respect to monthly net precipitation change in the Bakhtegan catchment" - P14, L2: "KB shows the relationship between net monthly precipitation and groundwater level variation":

This is true, and other reviewers also pointed out this point. (Page 8 – Line 3) must be corrected: (KB) shows the relationship between the groundwater volume changes and the precipitation. This will be corrected in the revised version.

* Specific Comments:

Specific comments (5 to 21) will be reviewed and corrected where needed in the revised version, as they are related to text structure

* Figure 6: What do the black lines surrounding the aquifers represent:

they represent the aquifers boundaries

―――――――――――――――――――

---

## Author Comment (AC4) · 15 Feb 2019

We want to thank our referee for his review and the insightful and comprehensive comments. Below is an aswer for the reviewer comments:

* How did the smaller study area affect the GRACE derived results:

The paper tends to discuss the usage of GRACE data in catchments smaller than the recommended area of 200,000 km2 suggested by (Longuevergne et al., 2010). Results (Figure 4B) showed that GRACE data proved the groundwater level depletion in the catchment. Still as the catchment area is considers small, around 16 percent of

the recommended area, the GRACE data was not totally fit with the in-situ collected data. In the discussion section (page 17 – line 7 till 18) we discuss more about this issue

* How does the geology of the catchment interrupt with the water mass variation:

The aquifer system found in the catchment are shallow alluvial aquifers. Deeper confined aquifers can be found lying beneath those alluvial aquifers. As groundwater level increase in depth, farmers are tending to dig dipper wells. The pressure stabilization in this cases are leading to a big drop in groundwater level. This is discussed more on (Page 14 Line 18 till Page 15 Line 2)

* How does the water mass change in the catchment indicates the groundwater recharge and depletion:

The water mass derived from the GRACE data shows the total monthly water change. This includes the groundwater volume variation, surface water, snow melt, soil moisture. . . Referring to equation 5 page 7, and as suggested by (Rodell and Famigliett 2002), the surface water volume and the soil moisture variation was removed from the water mass variation. (GWMI) is now the monthly groundwater variation by GRACE data

* What are other factors interrupting the water mass variation, other than discussed in the paper:

Most important factor that is the transboundary groundwater from other catchments. At smaller scale this might have an effects on the results obtained. Other than that we have discussed all the water balance components in the catchment